# *Alpha* and *Betacoronavirus* Detection in Neotropical Bats from Northeast Brazil Suggests Wide Geographical Distribution and Persistence in Natural Populations

**DOI:** 10.3390/ani15030332

**Published:** 2025-01-24

**Authors:** Thays Figueiroa, Marina Galvão Bueno, Patricia Emilia Bento Moura, Marcione Brito de Oliveira, José Luís Passos Cordeiro, Nádia Santos-Cavalcante, Giovanny A. Camacho Antevere Mazzarotto, Gabriel Luz Wallau, Leonardo Corrêa da Silva Junior, Paola Cristina Resende, Marilda M. Mendonça Siqueira, Maria Ogrzewalska

**Affiliations:** 1Fundação Oswaldo Cruz, IOC, Laboratório de Vírus Respiratórios, Exantemáticos, Enterovírus e Emergências Virais, Rio de Janeiro 21040-900, RJ, Brazil; thaysfigueiroa97@gmail.com (T.F.); leocorrra15@gmail.com (L.C.d.S.J.); paola@ioc.fiocruz.br (P.C.R.); mmsiq@ioc.fiocruz.br (M.M.M.S.); 2Fundação Oswaldo Cruz, IOC, Laboratório de Virologia Comparada e Ambiental, Rio de Janeiro 21040-900, RJ, Brazil; patiebm.m@gmail.com; 3Museu Nacional, Departamento de Vertebrados, Setor de Mastozoologia, Universidade Federal do Rio de Janeiro, Rio de Janeiro 20940-040, RJ, Brazil; oliveira01marcione@gmail.com; 4Fundação Oswaldo Cruz, Unidade do Ceará, Área de Saúde e Ambiente, Eusébio, Ceará 61773-270, CE, Brazil; jose.cordeiro@fiocruz.br; 5Plataforma Internacional para Ciência, Tecnologia e Inovação em Saúde (PICTIS), Via do Conhecimento, Edifício Central, 3830-352 Ílhavo, Portugal; 6Museu de História Natural do Ceará Prof. Dias da Rocha, Universidade Estadual do Ceará, Pacoti, Ceará 62770-000, CE, Brazil; nadiacavalcant@gmail.com; 7Fundação Oswaldo Cruz, Instituto Lêonidas and Maria Deane (ILDM), Unidade da Amazônia, Manaus 69057-070, AM, Brazil; 8Fundação Oswaldo Cruz, Unidade do Ceará, Laboratório Analítico de Competências Moleculares e Epidemiológicas, Plataforma de Camelídeos e Produção de Nanocorpos, Eusébio, Ceará 61773-270, CE, Brazil; giovanny.mazzarotto@fiocruz.br; 9Fundação Oswaldo Cruz, Departamento de Entomologia e Núcleo de Bioinformática, Instituto Aggeu Magalhães (IAM), Cidade Universitária, Recife 50740-465, PE, Brazil; gabriel.wallau@fiocruz.br; 10Department of Arbovirology and Entomology, Bernhard Nocht Institute for Tropical Medicine, WHO Collaborating Center for Arbovirus and Hemorrhagic Fever Reference and Research, National Reference Center for Tropical Infectious Diseases, 20359 Hamburg, Germany

**Keywords:** Chiroptera, zoonosis, respiratory viruses, South America, coronaviruses, RdRp

## Abstract

**Simple Summary:**

Our world is facing an urgent challenge with the rise of zoonotic viral diseases like COVID-19, emphasizing the critical need to understand viruses in wildlife. Bats, comprising about 20% of mammalian species globally, are known reservoirs for many pathogens, including coronaviruses. This study focused on investigating the presence and variety of coronaviruses in bat populations from northeastern Brazil, particularly in Ceará, where bat pathogen research is scarce. Over a year, we collected oral and rectal swabs from 298 bats across three municipalities. Our molecular analyses unveiled alphacoronaviruses in several bat species, and we identified a new type of *Betacoronavirus* in *Artibeus planirostris*, broadening our understanding of the coronavirus diversity in Brazilian bats. Furthermore, our findings suggest that closely related coronavirus strains can infect various bat species in distant Brazilian regions and habitats. While we did not find SARS-CoV-2 or influenza A viruses in the sampled bats, our study emphasizes the need for ongoing surveillance to detect and track zoonotic viruses circulating in wildlife populations, which is crucial for preventing future pandemics.

**Abstract:**

The emergence of zoonotic viral diseases, notably exemplified by the recent coronavirus disease pandemic in 2019 (COVID-19), underscores the critical need to understand the dynamics of viruses circulating in wildlife populations. This study aimed to investigate the diversity of coronaviruses in bat populations from northeastern Brazil, particularly in the state of Ceará, where little research on bat pathogens has been conducted previously. Bat sampling was performed between March 2021 and March 2022 across three municipalities, resulting in the collection of oral and rectal swabs from 298 captured individuals. Molecular analyses revealed alphacoronaviruses in multiple bat species. Additionally, a novel *Betacoronavirus* was identified in *Artibeus planirostris*, which did not fall within an established subgenus. Phylogenetic placement of these new coronavirus sequences suggests that closely related coronavirus lineages can infect a wide range of bat species sampled in distantly related Brazilian states and biomes. No SARS-CoV-2 and influenza A viruses were found in the sampled bats. These findings expand our understanding of coronavirus diversity in Brazilian bats. The detection of coronaviruses in various bat species underscores the importance of bats as reservoirs for these viruses. The absence of SARS-CoV-2 in the sampled bats indicates a lack of spillback events from human or environmental sources. However, the potential for future transmission events underscores the importance of ongoing surveillance and transmission mitigation protocols in wildlife management practices.

## 1. Introduction

Bats, as other vertebrate species (non-human primates, rodents, ungulates, and birds), are highly diverse and motile taxa of great relevance in the One Health context of multi-host pathogens. They represent approximately 20% of the global mammal population [1], comprising around 1474 known species [2]. Approximately 450 of these species are found in the neotropical region but they have a broad geographic distribution, with specimens found on all continents except Antarctica [3,4]. Additionally, bats exhibit a high diversity of diets [5] and generally display gregarious behavior, often forming large colonies either to promote thermoregulation or for protection against predators [6]. The flight adaptation in chiropterans has likely selected a distinct and innate immune response in these animals, associated with genes in the oxidative phosphorylation pathway responsible for supplying energy during sustained flight [7]. Furthermore, they are generally capable of expressing Toll-like receptors associated with the action of pro-inflammatory and antiviral cytokines, inhibiting viral replication or even inducing apoptosis of infected cells [8]. At the same time, the risk of a generalized inflammatory response due to the constitutive expression of interferon-alpha is neutralized through attenuation of NLRP3 inflammasome activation in primary immune cells. Thus, bats are well adapted to viral infections and are considered competent hosts or even reservoirs for different viral agents with the potential to threaten human health, as seen with Hendra, Nipah, Marburg, Rabies, and various coronaviruses, including those responsible for MERS, SARS, and probably SARS-CoV-2 [9,10,11,12].

Coronaviruses are single-stranded positive-sense RNA viruses with genomes of 16 to 31 kb [13]. Members of the family *Coronaviridae* (realm, *Riboviria*; kingdom, *Orthornavirae*; phylum, *Pisoniviricetes*; order, *Nidovirales*; suborder, *Cornidovirineae*) can infect a wide range of animals and are generally associated with respiratory and/or digestive illness [13,14]. Research into coronaviruses and the potential spillover from bat species has become more important since the 2002 epidemic of SARS-CoV, which was presumably transmitted from masked palm civets (*Paguma larvata*) to human populations, although its origin might be traced back mainly to horseshoe bats (*Rhinolophus* species) from Asia [12,15,16,17,18,19]. Subsequent studies unveiled a diverse range of novel bat coronaviruses from *Alpha* and *Betacoronavirus* genera circulating in bats worldwide [20], but their global diversity and distribution is non-random and driven by variation in the biogeography of bats, and their richness is estimated to be much higher than previously appreciated [21].

In South America, particularly in Brazil, there is a noticeable scarcity of studies on pathogens associated with wild animals, including bats and coronaviruses [22]. Notably, none of these studies have been conducted in the state of Ceará, located in the northeastern region of Brazil. The study area, Serra de Baturité, constitutes one of the most important enclaves of moist forest in Ceará [23]. The mountain forests within these areas are recognized as ecological disjunctions of the Atlantic Forest, making them remnants of high biodiversity [24]. Additionally, the regions are considered ecotourism sites with the potential for human interaction [25].

Considering this, we tested bats sampled in the state of Ceará for coronaviruses encompassing SARS-CoV-2 and other related coronavirus species, as well as for the influenza A virus. This initiative commenced in March 2021, over a year after the first autochthonous case of COVID-19 was detected in Brazil [26]. The aim was to shed light on the presence and potential transmission of these viruses among animal populations, contributing to a more comprehensive understanding of the dynamics and risks associated with respiratory pathogens in both wildlife and human contexts.

## 2. Materials and Methods

### 2.1. Study Area and Capture Sites

Bat captures were conducted between March 2021 and March 2022 in the northeast region of Brazil, which is predominantly characterized as semi-arid and primarily covered by caatinga vegetation [27]. Bat sampling was carried out in three municipalities in the state of Ceará, located in northeastern Brazil: Guaramiranga, Pacoti, and Eusébio. Within these municipalities, six sampling sites were selected. In Guaramiranga, the chosen sites were Sítio Nova Olinda (4°14′48.11″ S, 38°56′29.38″ W) and Sítio Riacho Fundo (4°15′40.37″ S, 38°55′5.41″ W). In Pacoti, sampling occurred at the headquarters of the Natural History Museum of Ceará Prof. Dias da Rocha (4°13’38.87″ S, 38°55’21.25″ W), Sítio São João (4°13′33.21″ S, 38°54′50.09″ W), and Sítio Boa Vista (4°13′6.85″ S, 38°54′7.00″ W). In Eusébio, a municipality that is part of the metropolitan region of Fortaleza, the capital of Ceará, sampling was conducted at the headquarters of Fiocruz Ceará, close to Lagoa da Precabura (3°50′11.89″ S, 38°26′41.93″ W) (Figure 1).

### 2.2. Bat Sampling

Each site was sampled for two consecutive nights using eight mist nets, measuring 9 × 3 m, installed at ground level in clearings, trails, near rivers, or residences. The nets were opened at 6:00 PM and maintained for 6 h, with checks every 15 min. Specimens captured were placed in cloth bags, sorted, and identified using keys available in the literature [28,29]. For each individual, the species name, date and time of capture, forearm length (mm), body mass (g), sex, and age category were recorded. Age category was determined by examining the epiphyses of the phalanges, classifying individuals as young, subadults, or adults [30]. The reproductive status of females was visually or palpably assessed, then categorized as inactive, pregnant, lactating, or post-lactating. Captured individuals without indications of trauma or stress, as well as pregnant or lactating females, were released after recording relevant data. Nomenclature followed [31]. The collected specimens were accessioned into the mammal collection of the Museu de História Natural do Ceará Prof. Dias da Rocha.

Oral and rectal swabs were collected from all captured individuals. Rectal samples were gathered using sterile Ultra-Thin Minitip Size Nylon^®^ Flocked Swabs (Copan Italia, Brescia, Italy), while oral swabs were employed with regular rayon swabs. All samples were collected and preserved in 1 mL RNAlater™ Stabilization Solution (Invitrogen™, Waltham, MA, USA).

### 2.3. Viral RNA Extraction and Virus Detection

To extract viral RNA, 140 µL of the sample was utilized. The QIAamp^®^ Viral RNA Mini Kit (Qiagen, Hilden, Germany) was used for the extraction process, following the manufacturer’s instructions. The elution buffer was used to elute the viral RNA, resulting in a final volume of 60 µL. The extracted RNA was promptly stored at −80 °C until molecular analysis. Negative controls, using RNAse/DNAse-free water, were included in each extraction procedure. Following the extraction, genetic material from the collected biological samples underwent analysis with RT-PCR, real-time RT-PCR, and partial genome sequencing to examine the selected viral panel.

For the detection of SARS-CoV-2 and influenza A virus in animals, a real-time RT-PCR assay was employed. The extracted RNA underwent screening through a SARS-CoV-2 and InfA (for all influenza A subtypes) real-time RT-PCR utilizing the SARS-CoV-2 detection Molecular InfA/E/RP Kit (Biomanguinhos, Rio de Janeiro, Brazil), following the methodology outlined by [32]. Both reverse transcription and amplification procedures were conducted without duplication on the ABI7500 platform (Thermo Fisher Scientific, Waltham, MA, USA). Samples exhibiting a cycle threshold (Ct) value below 38 were considered positive. Each run incorporated a negative control (RNAse/DNAse-free water) and positive control RNA templates (synthetized genblock provided by Molecular InfA/E/RP Kit) for validation and quality assurance.

Screening for other coronaviruses (CoVs) was conducted using conventional panco-ronavirus RT-PCR. All samples underwent pancoronavirus RT-PCR, specifically targeting the RNA-dependent RNA polymerase (RdRp) gene, following the procedure outlined by [33,34]. In brief, cDNA was synthesized and amplified in a first-round PCR using the One-Step RT-PCR Enzyme Mix Kit (Qiagen, Hilden, Germany). The primers used in the first round were RdRp S1 (5′-GGKTGGGAYTAYCCKAARTG-3′) and RdRp R1 (5′-TGYTGTSWRCARAAYTCRTG-3′) [32], generating an expected product size of 602 base pairs (bps). Reactions were conducted in a Veriti Thermo Cycler (Applied Biosystems, Waltham, MA, USA) with the following conditions: reverse transcription (50 °C, 30 min), reverse transcriptase inactivation, and DNA polymerase activation (95 °C, 15 min), followed by 40 cycles of DNA denaturation (94 °C, 45 s), annealing (52 °C, 45 s), and extension (72 °C, 45 s), concluding with a final extension step (72 °C, 10 min). Subsequently, nested PCR was carried out using the Phusion RT-PCR Enzyme Mix Kit (Sigma-Aldrich, San Luis, MO, EUA), along with primers Bat1F (5′-GGTTGGGACTATCCTAAGTGTGA-3′) and Bat1R (5′-CCATCATCAGATAGAATCATCAT-3′) [34]. One microliter of the amplified product from the first round was used as a template in the nested PCR, conducted under the following conditions: denaturation (98 °C, 30 s), followed by 35 cycles of DNA denaturation (98 °C, 15 s), annealing (52 °C, 15 s), extension (72 °C, 30 s), and a final extension step (72 °C, 5 min). The RdRp amplicons, approximately 440 bps in size, were visualized on 1.5% agarose gels using SYBR™ Safe DNA Gel Stain (Thermo Fisher Scientific, Waltham, MA, USA). The positive controls utilized for PCR consisted of RNA samples of SARS-CoV-2 extracted from an isolated virus, following the same procedures outlined for animal samples.

For Sanger sequencing, the DNA of RdRp amplicons (440 bps) was purified using the QIAquick Gel Extraction Kit (Qiagen, Hilden, Germany) following the manufacturer’s recommendations. The Sanger sequencing reaction was prepared utilizing the BigDye Terminator v3.1 Cycle Sequencing Kit (Life Technologies, Carlsbad, CA, EUA) with primers Bat1F and Bat1R at a concentration of 3.2 pmoles. Sequencing was executed on the ABI 3730 DNA Analyzer (Applied Biosystems, Waltham, MA, USA) following the protocols established by [35]. Reads were assembled and were evaluated using Sequencher 5.1 (GeneCodes, Ann Arbor, MI, USA) to obtain the final consensus, which was uploaded to the National Center for Biotechnology Information (NCBI) GenBank database.

### 2.4. Coronaviridae Sequence Data Recovery and Phylogenetic Analysis

To determine from which *Coronavirus* genus the obtained sequences belong to, we first performed the amino acid translation of the fragments obtained in this study using ORFinder (https://www.ncbi.nlm.nih.gov/orffinder/, accessed on 14 February 2023—any sense codon option) and aligned the amino acid sequence from the largest reading frame to the full RdRp region of reference *Coronaviridae* genomes (Appendix A) using MAFFT v7.0 [36] online (https://mafft.cbrc.jp/alignment/server/index.html, accessed on 14 February 2023) with default parameters. No trimming was performed. We then performed a maximum likelihood phylogenetic reconstruction using IQTREE 2.0 [37] using the GECV01031551.1_MLeV as an outgroup. Amino acid substitution models were selected according to the best model suggested by ModelFinder, then implemented within IQTREE 2.0 as reported in each figure legend.

To gain further insights and a higher phylogenetic understanding of the sequences generated in this study, we then performed homologous sequence retrieval from the non-redundant NCBI database based on BLAST searches of all partial RdRp obtained in this study. Up to the 100 first best hits were recovered based on the highly similar sequence option (megablast). Due to the high nucleotide divergence between *Coronavirus* genera at the nucleotide level, we performed a within genus partial RdRp nucleotide alignment and included one outgroup sequence from the most closely related genus (full trees can be accessed in the Appendix A). Nucleotide alignments were performed with MAFFT v7.0 online (https://mafft.cbrc.jp/alignment/server/index.html, accessed on 20 June 2023) with the --add option and --keeplength option based on the highly curated initial alignment of the reference genomes (reference genomes accession numbers can be found at Appendix A). In order to evaluate the phylogenetic clustering of the sequences at the subgenus level, we also included sequences with curated subgenus metadata associated with the ICTV (https://ictv.global/report/chapter/coronaviridae/coronaviridae/alphacoronavirus, https://ictv.global/report/chapter/coronaviridae/coronaviridae/betacoronavirus, accessed on 20 June 2023).

Maximum likelihood phylogenetic analyses were performed using IQTREE 2.0 [37] at the amino acid and nucleotide levels. Substitution models were selected with ModelFinder [38] and implemented within IQTREE. Branch support was evaluated with an approximate likelihood ratio test (aLRT) [39] and ultrafast bootstrap (UFBOOT = 1000) [40].

## 3. Results

A total of 298 bats from six sampling sites within three municipalities were analyzed in the present study (Table 1 and Appendix A). This diverse collection exhibited a rich array of biodiversity, encompassing 16 genera, 22 species, and five families: Emballonuridae, Molossidae, Noctilionidae, Phyllostomidae, and Vespertilionidae (Table 1 and Appendix A). The most frequently sampled bats in the present study were *Carollia perspicillata* (N = 76), *Glossophaga soricina* (N = 48), *Artibeus planirostris* (N = 37), *Phyllostomus discolor* (N = 33), and *Sturnira lilium* (N = 32). Most of the sampled species (19) were only represented by less than ten individuals (Table 1).

All oral and rectal swabs tested from these captured bats returned negative results for influenza A and SARS-CoV-2. However, 21 bats yielded positive results in RT-PCR for *Coronavirus* and after sequencing and comparison with the sequences available in GenBank, *Alphacoronavirus* RNA was detected in 20 (6.7%) of the tested bats, all of which were from the Phyllostomidae family, encompassing seven different species: *C. perspicillata* (7/76, 9.2%), *G. soricina* (3/48, 6.2%), *Lonchorhina aurita* (2/4, 50%), *Platyrrhinus lineatus* (1/9, 11.1%), *Platyrrhinus recifinus* (1/7, 14.2%), *P. discolor* (5/33, 15.1%), and *S. lilium* (1/32, 3.1%). In one male *A. planirostris* from the Pacoti municipality, a *Betacoronavirus* was detected (1/37, 3.7%). All positive samples except two were from rectal swabs (Table 2). High quality consensus sequences, based on the alignment of forward and reverse Sanger reads, with an expected size of 400–420 bps, were successfully obtained from 17 samples and deposited in GenBank (Table 2). None of the sampled bats exhibited any clinical signs of disease.

Phylogenetic reconstruction including the known diversity of *Coronavirus* genera confirmed that 16 out of 17 sequences were clustered with high branch support within the *Alphacoronavirus* genus (Figure 2A, Appendix A).

The single sequence from the *Betacoronavirus* genus detected in *A. planirostris* did not cluster within known *Betacoronavirus* subgenera (Figure 2A). At the higher resolution phylogenetic analysis, the novel *Betacoronavirus* sequence clustered with high node support to two other coronavirus sequences recovered from the *Artibeus* species (*A. planirostris* and *Artibeus lituratus*) sampled in 2011 and 2012, which are not members of known *Betacoronavirus* subgenera (Figure 2B, Appendix A).

Nine out of sixteen *Alphacoronavirus* sequences clustered into the *Amalacovirus* subgenus (Figure 3A). These sequences are grouped with other characterized viruses from *P. discolor*, sampled in different regions of Brazil (Pernambuco state also in Northeast Brazil) as well as in Colombia and Panamá (Figure 3A). On the other hand, five sequences from *C. perspicillata* (4) and *P. lineatus* (1) clustered in a monophyletic clade that is a sister to *Amalacovirus* subgenus, but lacks a subgenus classification (Figure 3B, Appendix A). These sequences clustered with high node support to other *Coronavirus* sequences recovered from Brazilian bats between 2012 and 2021, such as from other populations of *C. perspicillata* and *A. planirostris* (Figure 3B). One sequence from *S. lilium* clustered with other sequences recovered from different populations of the same species sampled between 2012 and 2021 in Brazil in a sister clade (with no known subgenus) of the subgenus *Tegacovirus* and *Minacovirus* (Figure 4A). Lastly, one sequence obtained from *P. recifinus* clustered with sequences from *A. lituratus* and *Artibeus jamaicensis* sampled in Brazil, Colombia, and Panama, forming a sister clade (with no known subgenus) of the subgenus *Decacovirus* (Figure 4B).

## 4. Discussion

The 298 bats analyzed belong to 22 species from the Emballonuridae, Molossidae, Noctilionidae, Phyllostomidae, and Vespertilionidae families. A total of 20 individuals from the Phyllostomidae family were found to be carrying several alphacoronaviruses from the *Amalacovirus* subgenus and alphacoronaviruses that clustered in monophyletic clades with no known subgenera, but were sister clades to the *Amalacovirus*, *Tegacovirus*, *Minacovirus*, and *Decacovirus* subgenera. One *A. planirostris* was found to be infected with a novel *Betacoronavirus*. This study represents the first detection and classification of coronaviruses from bats sampled in the Brazilian state Ceará.

In Brazil, Bat-CoVs have been identified in over 29 bat species spanning the Phyllostomidae, Molossidae, and Vespertilionidae families, exhibiting diverse dietary preferences, including omnivory, frugivory, nectarivory, insectivory, and hematophagy [41,42,43,44,45,46,47,48]. The 7% prevalence of CoVs we observed falls within the range reported in other studies of neotropical bats. Detection rates in these studies varied widely, from as low as 0–1% [49,50] to around 2.7–3.7% [42,45,51], reaching as high as 12.9–17.2% [43,48]. However, it is important to highlight that differences in detection rates can be attributed to many factors such as seasonality, increased numbers of certain species being included in different studies [41,46,47], or different types of samples tested [50,52].

Most Bat-CoVs sequences identified in Brazil fall within the *Alphacoronavirus* genus [41,42,43,45,46,47,48,52], a trend which is consistent with our own findings. Our data further reveal that the sequences generated in this study are linked to various distinct subclades of alphacoronaviruses. We observed that coronaviruses from hosts of the same species are closely related. For example, sequences of CoV (*Amalacovirus*) obtained from *P. discolor* were grouped with other viruses characterized from *P. discolor* samples recovered in different regions of South and Central America. Similarly, CoV recovered from *S. lilium* in the present study clustered with other sequences recovered from different populations of the same species sampled in southern Brazil (Sao Paulo state). However, some CoVs recovered here also clustered in the same clade with CoVs recovered from distantly related bat species belonging to different genera. For instance, *Amalacovirus* from *P. discolor* clustered together with CoVs recovered in *C. perspicillata* and *S. lilium*. In another case, CoVs found in *C. perspicillata* were closely related to the CoVs detected in *P. lineatus*. These bat genera visit the same fruits, which could facilitate viral cross-species transmission. Although sporadic observations of closely related coronaviruses in different bat species and even families have been reported among neotropical bats [43,48,53], it appears that host-switching events are less common in Latin America compared to bats in Africa and Asia [21]. It was suggested that coronaviruses sampled from Latin American bats still undergo host-switching, but they tend to move between closely related species whereas, in Africa and Asia, viruses are more likely to cross the species barrier between more distantly related species [21]. The mechanism underlying these differences is not yet fully understood, but it may reflect regional variations in the risk of disease emergence. This could be one of the factors contributing to the emergence of highly pathogenic coronaviruses such as MERS-CoV and SARS-CoV in Asia rather than in South America [21].

Only a few sequences from the *Betacoronavirus* genus have been detected in Brazil [44,45]. Thus, our most important finding is the *Betacoronavirus* genus detected in *A. planirostris*. This CoV did not cluster within a known *Betacoronavirus* subgenus and at the higher resolution phylogenetic analysis, it clustered with high node support to other two coronavirus sequences within a non-defined subgenus recovered from the *Artibeus* species (*A. planirostris* and *A. lituratus*) sampled over decade ago in the Atlantic rainforest, southern Brazil.

Regions with higher bat diversity are likely to harbor a greater variety of viruses [21]. Therefore, it is anticipated that Brazilian bats, with their rich diversity, host a high diversity of coronaviruses. However, our current understanding remains limited, and ongoing research is expected to uncover more coronaviruses [22]. It has been estimated that sampling up to 400 individuals from each bat species would be necessary to capture the full diversity of coronaviruses [21]. Therefore, increased temporal and population sampling is warranted to further uncover more divergent coronaviruses in Brazil.

The higher prevalence of positive rectal samples compared to oral swabs is consistent with the findings of previous studies [12,21,53,54,55]. While feces serve as the primary source for CoV detection in bats, it is worth noting that detection in oral swabs has also been demonstrated, although less frequently [53,56]. Additionally, the study by [43], which identified CoVs not only in the intestines but also in the lungs and liver, underscores the multiple organ/tissue tropism of coronaviruses in bats, as has already been observed for other mammal species [57].

The captured bats showed no visible signs of disease, including those infected with CoVs, which is in line with the findings from previous studies [49,55,58]. This observation suggests that, despite the presence of CoVs within bat populations, individual bats may only experience transient infections with CoVs without displaying noticeable illness. In another study, it was noted that no bat remained persistently positive for CoV RNA after six weeks [59], providing additional evidence that CoV infections in bats may have a short duration and be self-limiting. This corresponds to the concept that bats have evolved mechanisms to coexist with coronaviruses, and these viruses may establish a more symbiotic relationship with their natural hosts compared to the severe outcomes observed in other species, such as humans.

The urgency and amplification of coronavirus research have significantly intensified since the emergence of the COVID-19 pandemic, attributed to the SARS-CoV-2 virus. Analogous to its predecessor, SARS-CoV, SARS-CoV-2 was presumed to have originated in *Rhinolophus* spp. in Asia [60,61,62], although the exact spillover mechanism remains yet undetermined. SARS-CoV-2 presents a potential threat to New World bat populations if bats encounter the virus through interspecies contact, if the virus can subsequently establish infection and propagate within bat populations, or if it engenders morbidity or mortality in bats. Furthermore, the establishment of SARS-CoV-2 within bat reservoirs could serve as a reservoir for secondary transmission to humans, domestic animals, or other wildlife species [63,64,65]. However, at present, SARS-CoV-2 has not been detected in wild New World bat populations, suggesting a lack of spillover from human sources [65,66], consistent with the outcomes of our investigations, which indicate that the virus is either not present or that it persists in undetectable levels in bat populations in this region. These findings offer encouraging prospects from both conservation and public health standpoints. Nonetheless, spillover events are contingent upon the level of exposure, and wildlife practitioners engaged in bat-related activities are urged to adhere strictly to transmission mitigation protocols [67,68].

Brazil does not harbor Old World Rhinolophidae and Hipposideridae bats, which are natural reservoirs of SARS-related coronaviruses. Consequently, the absence of SARS-CoV-2 in phyllostomid, molossid, and vespertilionid bats in the present study aligns with expectations, although it cannot be discarded in the future. It is also known that susceptibility to SARS-like viruses varies among bat species and families. SARS-like betacoronaviruses have been identified not only in horseshoe bats, but also in molossid bats (*Chaerephon plicatus*) in China [69] and in guano samples from vespertilionid and molossid bats in Europe [70]. Moreover, experimental studies involving neotropical bat species, such as *Eptesicus fuscus* (Vespertilionidae) [71] and the *Tadarida brasiliensis* (Molossidae) [72], suggest that these species exhibit minimal competence as hosts for SARS-CoV-2.

In the current investigation, we did not detect influenza A viruses. Influenza viruses have been identified in terrestrial, flying, and aquatic mammals, as well as in a wide variety of wild aquatic birds, which are considered natural reservoirs for avian influenza viruses (AIVs) [73]. Ten years ago, the new subtypes H17N10 and H18N11 were observed in fruit bats in Guatemala, Peru [74,75] and, more recently, in Brazil [76]. These discoveries suggest that neotropical bats may serve as a potentially significant and likely ancient reservoir for a diverse range of influenza viruses. However, these avian-associated influenza A viruses, which are phylogenetically highly divergent in their hemagglutinin (HA) and neuraminidase (NA) genes, have thus far only been detected in six individual bat specimens from the *S. lilium* (N = 4) and *A. planirostris* (N = 2) species [74,75,76]. The absence of this virus in our study may be attributed to its low prevalence, as observed in previous studies on neotropical bats (0% [50], 0.4% [76], 0.9% [75], and 1.0% [74]). Moreover, in 2017, a novel IAV A(H9N2)-like virus was isolated from Egyptian fruit bats (*Rousettus aegyptiacus*) [77] that has features associated with an increased risk to humans. This emphasizes the need for continued surveillance of influenza A among South American bats [77,78].

Bat fauna sampling was conducted using mist nets, a method commonly employed in the neotropical region. Although they are efficient in capturing bats of the Phyllostomidae family, especially frugivorous species, their effectiveness is reduced for insectivorous bats as these species tend to fly at higher altitudes and avoid mist nets by using echolocation [79,80]. However, the recent documentation of the first sightings of *Myotis ruber* and *Molossus pretiosus* species in the same region as the present study, Ceará [81], captured using mist nets, plays a crucial role in expanding our knowledge of bat biodiversity. These new records represent a valuable contribution to the list of known species in the region [82], providing a more robust foundation for future research and conservation initiatives aimed at protecting bat diversity in the region. Therefore, this study can contribute to the understanding of bat biodiversity and offer valuable insights for research related to the ecology and epidemiology of diseases with the potential for transmission by bats.

We are aware of the limitations of our study. Ideally, isolating the virus would have been optimal; however, due to the remote locations where samples were collected, preservation in RNAlater™ Stabilization Solution was necessary, precluding attempts to isolate the virus. Another limitation is that some species were not well represented in our sample set through the use mist nets for sampling, as discussed above. Additionally, we only obtained short fragments of the genome, which limited our analyses. Ideally, having the complete genome would provide a more comprehensive understanding of the viruses present.

## 5. Conclusions

Our study provides valuable insights into the prevalence and diversity of coronaviruses in bat populations in northeastern Brazil. The detection of alphacoronaviruses in various bat species underscores the importance of bats as reservoirs for these viruses. Additionally, a single *Betacoronavirus* was detected in *A. planirostris* which does not fall within established genera. Higher resolution phylogenetic analysis indicated that it is closely related to two other coronavirus sequences which had been recovered from *A. planirostris* and *A. lituratus* bats captured in southern Brazil over a decade ago. The absence of SARS-CoV-2 in the sampled bats indicates a lack of spillback events from human or environmental sources. However, the potential for future transmission events underscores the importance of ongoing surveillance and transmission mitigation protocols in wildlife management practices. Continued research into bat pathogens and their interactions with humans and other animals is essential for developing comprehensive strategies to promote public health. Understanding these dynamics is crucial for effectively mitigating the risks of zoonotic diseases.

## Figures and Tables

**Figure 1 animals-15-00332-f001:**
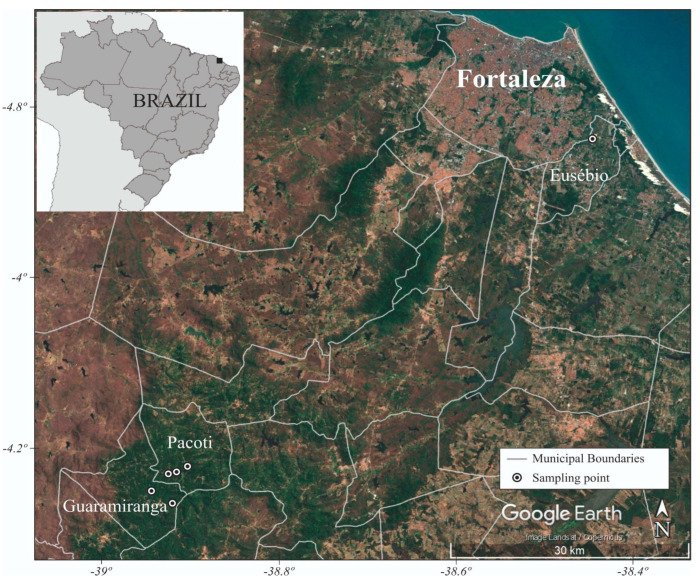
Sampling points of bats in the state of Ceará, located in Guaramiranga, Pacoti, and Eusébio municipalities.

**Figure 2 animals-15-00332-f002:**
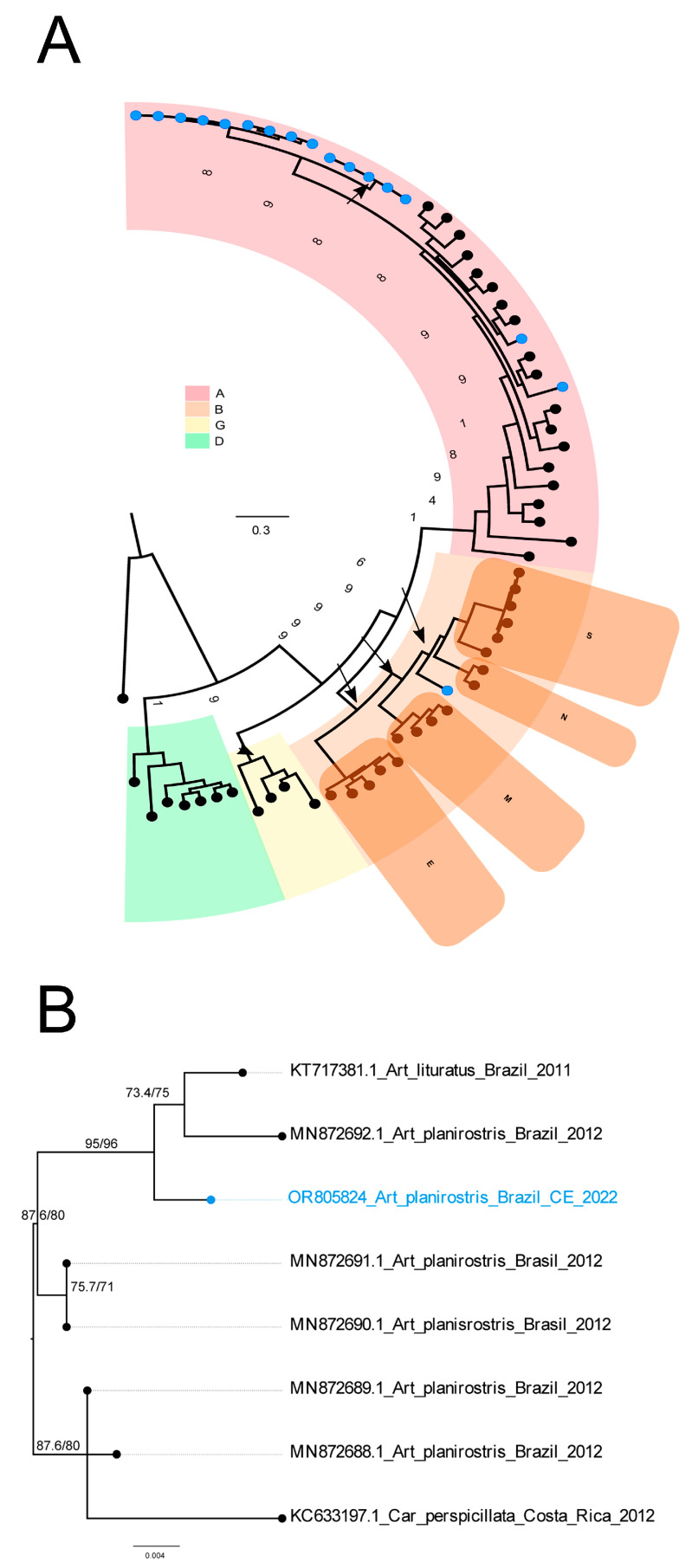
Maximum likelihood phylogenetic reconstructions with reference genomes covering the known diversity of genera from the *Coronaviridae* family. (**A**) Phylogenetic tree reconstructed based on the full RdRp amino acid sequences of reference *Coronavirus* genomes plus the RdRp fragments recovered in this study using the best amino acid substitution model Q.insect+I+I+R5-, *Coronavirus* genera coded by color and the subgenera of the genus *Betacoronavirus*. Tips with blue circles denote the sequences generated in this study (partial RdRp). Branch support of key nodes are shown (aLTR/UFBOOT). (**B**) Pruned tree of the nucleotide-based phylogenetic reconstruction of the *Betacoronavirus* genus including RdRp partial sequences from the reference genomes (part A) and 440 bps of the 70 best blast hits against the non-redundant NCBI database, plus the sequence recovered in this study using the nucleotide substitution model GTR+F+I+G4. Blue tip and name denote the nucleotide sequence generated in this study, branch lengths are amino acid and nucleotide substitutions per site. Tip name abbreviations are related to the bat genus as follows: Art, *Artibeus*; Car, *Carollia*. Nexus files in the Appendix A.

**Figure 3 animals-15-00332-f003:**
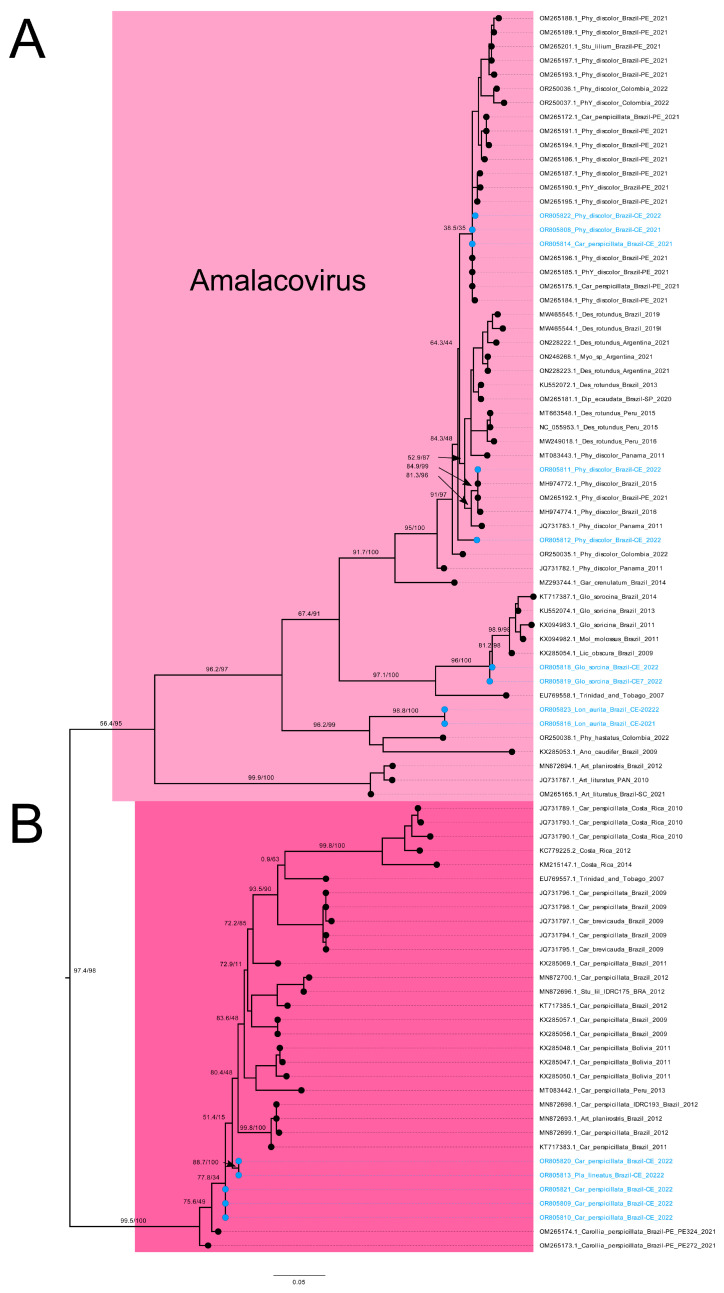
Maximum likelihood pruned trees focusing on *Amalacovirus* and a sister non defined subgenus of the whole *Alphacoronavirus* tree (Appendix A) reconstructed using the best nucleotide substitution model GTR+F+I+I+R5. (**A**)—Phylogenetic tree excerpt showing nine sequences obtained in this study clustered within the *Amalacovirus* subgenus and (**B**)—a sister monophyletic clade with five sequences with no known *Alphacoronavirus* subgenus (dark pink clade). Branch lengths are nucleotide substitutions per site. Art, *Artibeus*; Car, *Carollia*; Phy, *Phyllostomus*; Stu, *Sturnira*; Des, *Desmodus*; Myo, *Myotis*; Dip, *Diphylla*; Glo, Glossophaga; Lic, *Lichonycteris*; Lon, *Lonchorhina*; Ano, *Anoura*; Pla, *Platyrrhinus*; Mol, *Molossus*; Gar, *Gardnerycteris*. The highlighted in blue circles represent the sequences generated in the present study.

**Figure 4 animals-15-00332-f004:**
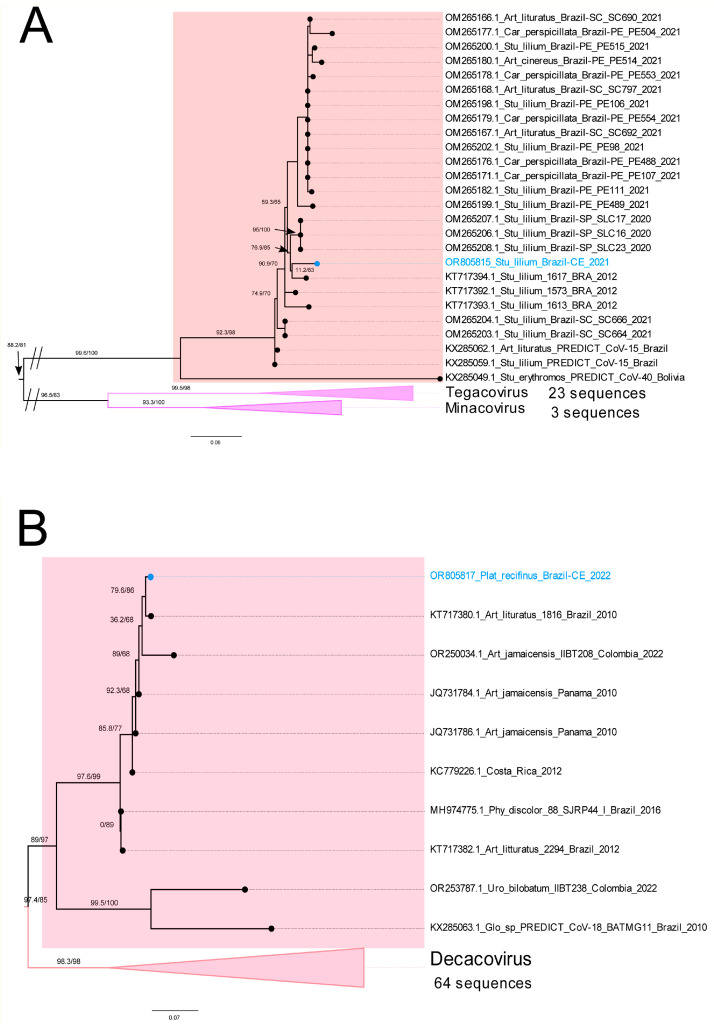
Maximum likelihood pruned trees of non-defined subgenera. These genera are phylogenetically related to the subgenera *Tegacovirus, Minacovirus*, and *Decacovirus* from the whole *Alphacoronavirus* tree (Appendix A) reconstructed using the best nucleotide substitution model GTR+F+I+I+R5. (**A**) A single sequence obtained in this study showing well-supported clustering into another clade with no known *Alphacoronavirus* subgenus association (pinkish square). (**B**) Phylogenetic tree excerpt showing a sequence obtained in this study clustered within a monophyletic clade with no known *Alphacoronavirus* subgenus associated, which is a sister clade of *Decacovirus* subgenus. Branch support is aLRT/UFBOOT, sequences obtained in this study are highlighted in blue, and branch lengths are nucleotide substitutions per site. Art, *Artibeus*; Car, *Carollia*; Stu, *Sturnira*; Glo, *Glossophaga*; Pla, *Platyrrhinus*; Phy, *Phyllostomus*; Uro, *Uroderma*.

**Table 1 animals-15-00332-t001:** Bats infected with CoVs across three municipalities in Ceará, Brazil.

Family	Species	No of Bats Captured/No of Bats Infected with CoVs					
		Eusebio		Guaramiranga		Pacoti	
		Male	Female	Male	Female	Male	Female
Emballonuridae	*Saccopteryx leptura*	0/0	0/0	0/0	0/0	0/0	1/0
Noctilionidae	*Noctilio leporinus*	0/0	3/0	3/0	1/0	0/0	0/0
Molossidae	*Molossus molossus*	1/0	1/0	0/0	0/0	0/0	1/0
Phyllostomidae	*Artibeus cinereus*	1/0	1/0	3/0	2/0	3/0	4/0
	*Artibeus fimbriatus*	0/0	0/0	0/0	0/0	1/0	0/0
	*Artibeus lituratus*	1/0	1/0	0/0	0/0	0/0	0/0
	*Artibeus obscurus*	0/0	0/0	0/0	0/0	0/0	1/0
	*Artibeus planirostris*	11/0	14/0	0/0	0/0	10/1	2/0
	*Artibeus* spp.	0/0	0/0	0/0	1/0	1/0	1/0
	*Carollia perspicillata*	4/0	2/0	15/2	13/2	28/1	14/2
	*Glossophaga soricina*	2/0	1/0	0/0	0/0	21/1	24/2
	*Lonchophylla mordax*	0/0	0/0	0/0	0/0	1/0	0/0
	*Lonchorhina aurita*	0/0	0/0	0/0	0/0	2/0	2/2
	*Lophostoma brasiliense*	0/0	0/0	0/0	0/0	0/0	1/0
	*Micronycteris minuta*	0/0	0/0	0/0	0/0	1/0	0/0
	*Phyllostomus discolor*	2/1	5/0	4/1	6/1	9/1	7/1
	*Phyllostomus hastatus*	1/0	1/0	0/0	0/0	0/0	0/0
	*Phyllostomus* spp.	0/0	0/0	1/0	1/0	0/0	1/0
	*Platyrrhinus lineatus*	1/0	0/0	2/1	4/0	2/0	0/0
	*Platyrrhinus recifinus*	0/0	0/0	1/0	1/0	3/0	2/1
	*Sturnira lilium*	4/0	3/0	3/0	2/0	13/1	7/0
	*Sturnira tildae*	0/0	1/0	0/0	0/0	2/0	0/0
Vespertilionidae	*Eptesicus furinalis*	0/0	0/0	0/0	0/0	1/0	1/0
	*Myotis lavali*	1/0	2/0	0/0	0/0	1/0	2/0
	*Myotis* spp.	0/0	0/0	2/0	0/0	1/0	0/0
	Total	30/1	32/0	34/4	31/3	100/5	71/8

**Table 2 animals-15-00332-t002:** Positive samples for CoVs from bats collected in the state of Ceará, Brazil, and analyzed in the present study.

Bat ID	Sample Type	Locality	Bat Species	Sex	CoV	Virus Name	GenBank Accession Number
MBO477	R ^1^	Eusebio	*Phyllostomus discolor*	♂	Alfa	BatCoV/Phyllostomus_discolor/Brazil/CE/FIOCRUZ-A211186/2021	OR805808
NAC14	R	Guaramiranga	*Carollia perspicillata*	♀	Alfa	BatCoV/Carollia_perspicillata/Brazil/CE/FIOCRUZ-A220424/2022	OR805809
NAC28	R	Guaramiranga	*Carollia perspicillata*	♀	Alfa	BatCoV/Carollia_perspicillata/Brazil/CE/FIOCRUZ-A220428/2022	OR805810
PC118	R	Guaramiranga	*Phyllostomus discolor*	♀	Alfa	BatCoV/Phyllostomus_discolor/Brazil/CE/FIOCRUZ-A220484/2022	OR805811
PC122	R	Guaramiranga	*Phyllostomus discolor*	♂	Alfa	BatCoV/Phyllostomus_discolor/Brazil/CE/FIOCRUZ-A220488/2022	OR805812
ACA102	R	Guaramiranga	*Platyrrhinus lineatus*	♂	Alfa	BatCoV/Platyrrhinus_lineatus/Brazil/CE/FIOCRUZ-A220364/2022	OR805813
NAC36	R	Guaramiranga	*Carollia perspicillata*	♂	Alfa	Low quality sequence (279 bps)	not deposited
NAC32	O ^2^	Guaramiranga	*Carollia perspicillata*	♂	Alfa	Low quality sequence (157 bps)	not deposited
MBO497	R	Pacoti	*Carollia perspicillata*	♂	Alfa	BatCoV/Carollia_perspicillata/Brazil/CE/FIOCRUZ-A211329/2021	OR805814
PC19	R	Pacoti	*Sturnira lilium*	♂	Alfa	BatCoV/Sturnira_lilium/Brazil/CE/FIOCRUZ-A211383/2021	OR805815
MBO514	R	Pacoti	*Lonchorhina aurita*	♀	Alfa	BatCoV/Lonchorhina_aurita/Brazil/CE/FIOCRUZ-A211491/2021	OR805816
PC108	R	Pacoti	*Platyrrhinus recifinus*	♀	Alfa	BatCoV/Platyrrhinus_recifinus/Brazil/CE/FIOCRUZ-A220474/2022	OR805817
PC112	R	Pacoti	*Glossophaga soricina*	♂	Alfa	BatCoV/Glossophaga_sorcina/Brazil/CE/FIOCRUZ-A220478/2022	OR805818
PC156	R	Pacoti	*Glossophaga soricina*	♀	Alfa	BatCoV/Glossophaga_sorcina/Brazil/CE/FIOCRUZ-A220507/2022	OR805819
PC158	R	Pacoti	*Carollia perspicillata*	♀	Alfa	BatCoV/Carollia_perspicillata/Brazil/CE/FIOCRUZ-A220509/2022	OR805820
PC68	R	Pacoti	*Carollia perspicillata*	♀	Alfa	BatCoV/Carollia_perspicillata/Brazil/CE/FIOCRUZ-A220513/2022	OR805821
PC111	R	Pacoti	*Phyllostomus discolor*	♂	Alfa	BatCoV/Phyllostomus discolor/Brazil/CE/FIOCRUZ-A220477/2022	OR805822
NAC60	R	Pacoti	*Lonchorhina aurita*	♀	Alfa	BatCoV/Lonchorhina_aurita/Brazil/CE/FIOCRUZ-A220460/2022	OR805823
	O						
NAC39	R	Pacoti	*Artibeus planirostris*	♂	Beta	BatCoV/Artibeus_planirostris/Brazil/CE/FIOCRUZ-A220439/2022	OR805824
ACA49	R	Pacoti	*Phyllostomus discolor*	♀	Alfa	Low quality sequence (207 bps)	not deposited
PC05	R	Pacoti	*Glossophaga soricina*	♀	Alfa	Low quality sequence (170 bps)	not deposited

^1^ R—rectal swab, ^2^ O—oral swab.

## Data Availability

Sequence data have been deposited in GenBank, records are as follows: OR805808–OR805824.

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
