# Peer review of "Alpha and Betacoronavirus Detection in Neotropical Bats from Northeast Brazil Suggests Wide Geographical Distribution and Persistence in Natural Populations"

_animals, 2025, doi:10.3390/ani15030332_

Round 1
Reviewer 1 Report
Comments and Suggestions for Authors
In this research article, the authors describe the identification and classification of coronaviruses in close to 300 bats captured in Northeastern Brazil, a region from which few Bat CoV’s have previously been characterized. The authors investigate CoVs in the bats using a combination of PCR assays, partial RdRp sequencing, and phylogenetic analyses, finding that ~7% of the bats harbour a CoV. RdRp phylogenies revealed that all but one CoV are alphacoronaviruses. The remainder is a novel betacoronavirus RdRp sequence distinct from established subgenera. While the viral sequences and taxonomic insights generated by the authors are important and valuable, there are several recommended modifications to enhance the presentation and validity of their results. Individual names in the phylogenies are way too small to read, and the naming is difficult to interpret and follow.
See attachment for detailed review comments

None
Author Response
Reviewer comment: In this research article, the authors describe the identification and classification of coronaviruses in close to 300 bats captured in Northeastern Brazil, a region from which few Bat CoV’s have previously been characterized. The authors investigate CoVs in the bats using a combination of PCR assays, partial RdRp sequencing, and phylogenetic analyses, finding that ~7% of the bats harbour a CoV. RdRp phylogenies revealed that all but one CoV are alphacoronaviruses. The remainder is a novel betacoronavirus RdRp sequence distinct from established subgenera. While the viral sequences and taxonomic insights generated by the authors are important and valuable, there are several recommended modifications to enhance the presentation and validity of their results. Individual names in the phylogenies are way too small to read, and the naming is difficult to interpret and follow.
Response: Thank you for all your recommendations. We provided new phylogenetic trees with bigger letters.
Abstract Reviewer comment:Line 52/53: “a novel betacoronavirus genus was identified” may be an overstatement. Instead could say “a novel betacoronavirus was…”, or inform the reader that phylogenetic analysis placed a partial RdRp sequence of one viral sample confidently on a unique branch within a clade containing subgenera of betacoronavirus.
Response: We agree. Corrected to ‘novel betacoronavirus’.
Reviewer comment: Line 57/58: The conclusions in the abstract could be further developed, i.e., why and how do the findings highlight the importance?
Response: We made the conclusion longer as requested.
Reviewer comment:Line 59: “RdRp” could be an additional keyword.
Response: suggestion accepted
Introduction Reviewer comment: Lines 82-91: The relevance of this paragraph to the study could be further developed. The information contained in last sentence could be moved earlier in the paragraph to tell the reader why the nuances of bat immune responses make bats particularly important reservoirs for viruses. The specific details of immune function (Lines 84-89) represent general immune functions; it is a stretch to suspect that bats are somehow special in viral carriage – for context, humans also carry many viruses asymptomatically for long periods.
Response: We modified the whole introduction as requested by other reviewers.
Reviewer comment: Line 98: ‘the invaluable ecological services provided by bats’ is an underdeveloped idea. So is “environmental conservation” related to viral surveillance in bats. These points are presented at an awkward time in the introduction: the relevance of viral surveillance in bats to human health is clear, but not so much the relation of viral surveillance in bats to conservation. Recommend further developing the themes, or removing these points entirely.
Response: We agree. We removed the part of conservation.
Reviewer comment: Methods
Line 141/142: The volume of RNAlater Stabilization Solution should be included, so that the proportion of the sample used in extraction can be assessed (i.e., on Line 143, 140 microliters are used of how many microliters total?)
Response: it was 1 mL for all samples.. We add this information.
Reviewer comment: Line 158/159: the source of positive-control templates could be included, along with their storage solution, and whether or not they received equivalent treatment to the test samples (extraction with QIAmp kit).
Response: We add this information.
Reviewer comment: Line 151-159: Were real time RT-PCR reactions conducted with replication? This detail is missing.
Response: No. The reaction was conducted only once. We add this information.
Reviewer comment: Lines 189-: Alignments methods are critical in phylogenetics; need additional details.4
Response: Thank you for raising this point. We improved the material and methods section acoordigly.
Reviewer comment: Line 191: How was reading frame established when translating partial RdRp? Were all possible reading frames tested?
Response: We described in more details how these procedures took place in the new MS version. We used ORFinder to extract the amino acid sequences from the longer and continuous open reading once this RdRp amplified and sequenced region is not expected to have stop codons or frame shifts. After that amino acid translation took place to perform the analysis for Figure 2A.
Reviewer comment: Line 191: Similarly, how were reference sequence handled, and were they trimmed to match the partial RdRp? These details could impact the quality of the alignment, and thus the validity of the phylogenetic analysis.
Response: Thank you for raising this point, we clarified these points in the new version of the manuscript. First, we used complete RdRp amino acid sequences of reference genomes plus the amino acid sequences recovered from the 440bp fragments generated in this study. There are some recent studies showing that combining longer and shorter sequences can be beneficial to recover more stable phylogenetic trees and positionate shorter fragments into reference trees (https://www.nature.com/articles/s41467-023-39847-x). We used complete and partial aa sequences and were able to recover the expected phylogenetic relationship within the Coronaviridae family and were able to positionate the new sequences within this context. Lastly, for Figure 2B and Figure 3, all were recovered from 440bp nucleotide sequences including the ones extracted from reference genomes.
Reviewer comment: Line 194: “sequence recovery” is an unclear way to phrase the collection of sequences for inclusion in the secondary phylogenetic analysis.
Response: We rephrase this sentence to increase clarity.
Reviewer comment: Line 200: “online”, need url. Default paramenters?
Response: Yes, we used default parameters. We added the url in the MS text.
Reviewer comment: Line 202: Was IQTREE run locally? If so, newer versions have been released (e.g., 2.2 was released over a year ago). Recommend double checking version number used.
Response: Thank you for raising this point. Yes, we have run a local version of IQTREE. We rerun the analysis with a newer version and did not detect differences in the phylogenetic reconstruction.
Reviewer comment: Line 205: the number of UFBOOT tests performed should be indicated.
Response Added
Reviewer comment: Results Line 207: “Three distinct locations” may refer to the three localities in which sampling took place, but there are 6 sampling locations indicated in Figure 1. Two of the sampling sites in Pacoti look to be nearly as far apart as one of the Pacoti sites (left-most in Figure 1) is to the northernmost Guaraminga sites. Recommend revisiting the statement that bats were samples from three locations.
Response: We sampled three municipalities and their sampling sites. The information is correct. We modified the materials and methods to make it clearer about the three municipalities and six sampling sites
Reviewer comment: Line 216 (Table 1): Recommend reformatting the width of columns in this table (can’t trust publisher to do it). The width of the column Guaramiranga in the Table is wider than for the other two localities, for unclear reasons. Values for bat infections are bolded; this should be indicated in table sub-captions. Recommend that number of bats infected with a CoV could be combined in a single cell with total bats samples from that locality. E.g., for A. planirostris, the one male of ten total captured in Pacoti with a CoV could be indicated as 10 (1) or 10/1 with a diagonally split cell.
Response: Thank you. We reformatted Table 1 as recommended.
Reviewer comment: Line 227: Does “consensus sequences” refer to alignment of Forward and Reverse Sanger reads? Recommend reminding the reader how amplicon sequencing was performed. Also, briefly explain (in Methods?) why 400 bp served as a cutoff for ‘high-quality’ sequences.
Response: We add the information that consensus sequences refer to Forward and Reverse Sanger reads; We add the information that 440 bp was an expected size of our product.
Reviewer comment: Figure 2 (page 9): Figure 2A statement and text fails to indicate whether this is a nucleotide or amino acid-based tree. The model used should be indicated. The outgroup does not include a label, thus it is difficult to assess whether it is appropriate. The unit for the branch length scale is not included. Spelling error in legend (gammacoronavirus). Text for all labels is too small, and quality of figures can be improved. For Figure 2B: how was the root chosen? Recommend increasing label size for leaves/taxa and branch supports.
Response: Thank you for raising these points. We added the information about amino acid or nucleotide sequences used for phylogenetic reconstruction, the amino acid and nucleotide substitution model used and the branch length scale in the figure legend. Regarding the outgroup for the analysis shown in Figure 2A we have used the sequence GECV01031551.1_MLeV. We added such information in the material and methods section. Lastly, we agree with the reviewer and generated new figures with increased tip label font size. The root was chosen following what is described in the material and methods section, that is, using one or few sequences from sister Coronavirdae genera. For instance, for the Alphacoronavirus tree reconstruction we included sequences of SARS-CoV-2 (which belong to Betacoronavirus genus) as outgroups. While for the Betacoronavirus tree reconstruction we added the AF304460.1 299E Alphacoronavirus as an outgroup.
Reviewer comment: Lines 246-248: Unclear how 2B relates to 2A. The figure legend indicates that 2B is a pruned portion of a larger tree; which larger tree? Where do the branches in 2B resolve in 2A? I higher resolution phylogeny that can clearly demonstrate how the authors selected the nodes that define each sub-genus is important for developing the argument that the new betacoronavirus sequence does not fall within established sub-genera.
Response: Thank you for raising this point. First, it is important to clarify that Figure 2B, 3A and B and Figure 4 A and B are not directly related to Figure 2A. The phylogenetic tree shown in Figure 2 A is a more general tree showing the phylogenetic positioning of the sequences generated in this study within the major Coronaviridae genera using only a handful of reference genomes (Table 1S). On the other hand, the remaining phylogenetic trees have been built only considering one genus at the time, including reference sequences from Figure 2A for specific genera and some outgroups. We used the sequences generated in this study to recover the most similar sequences from the nr database of NCBI and then built a phylogenetic hypothesis based on nucleotide sequences to have a higher resolution of the phylogenetic clustering of the sequences within the body of sequences already available in the literature. Regarding the selection of nodes from the higher resolution trees we selected well supported nodes that allowed us to clearly see the positioning of the sequences generated in this study relative to what was available in the literature. The reviewer can access the full phylogenetic tree in the supplementary material. We extracted the known sub genera of Alpha and Betacoroanvirus from ICTV (https://ictv.global/report/chapter/coronaviridae/coronaviridae/alphacoronavirus, https://ictv.global/report/chapter/coronaviridae/coronaviridae/betacoronavirus) once we assumed that it is the most systematically revised and up to date information available. We then included reference elements with associated subgenus associated information within the alignment of the high resolution analysis. We added such information in the Material and Methods section of the manuscript.
Reviewer comment: Line 255: reader is directed to Figure 3B, but it appears as though the direction should be to 2B.
Response: Thank you, we corrected this point within the text.
Reviewer comment: Line 256-259: some species names are not italicize; please fix throughout
Response: Corrected.
Reviewer comment: Line 260: ‘sister monophyletic clade’ is unclear without directly stating what the clade is ‘sister’ to. Recommend rephrasing that the 5 partial RdRp sequences resolved in a monophyletic clade with other alphacoronaviruses that is sister clade to sub-genus Amalacovirus but lacks a subgenus designation itself.
Response: We adjusted accordingly to the reviewer's suggestion.
Reviewer comment: Figure 3 (page 11): insufficient image quality to read labels. Difficult to assess this portion of the results/paper as unable to confidently make out taxa on tree. No unit is provided for the branch length scales across A, B, or C. Circles at branch tips that correspond to sequences from this study should be coloured blue as well (achieving consistency with Figure 2). The number of individuals represented in each collapsed clade (Tegacovirus, Minacovirus, and Decacovirus) should be indicated.
Response: Thank you for raising this point. We generated new figures with increased tip label names, added the number of sequences embedded into collapsed clade. We also added the branch length unity in the figure legend.
Reviewer comment: Line 272 (Figure 3): the entire tree from which these figures were pruned could be included as a supplement.
Response: Thank you, we agree with the reviewer and added the full trees as supplementary material of the manuscript.
Reviewer comment: Line 273: The supplementary material does not correspond to the number of sequences included in this tree. For instance, there is only one member of subgenus Amalacovirus in the supplement, despite there being many in Figure 3A.
Response: As supplementary material Table 1S we provided the accession number of the reference genomes used in Figure 2A, but now we included the full phylogenetic trees where the readers can access the accession number of all sequences recovered from databases for each particular analysis.
Reviewer comment: Discussion and conclusions. Recommend beginning the discussion with a recap of the most important results, and reasons for their significance.
Response: Thank you for your suggestion. We made a recap.
Reviewer comment: Lines 283-290: this is background information on viral surveillance in bats, that may fit better in the introduction.
Response: As recommended we removed this part to the introduction.
Reviewer comment: Line 294: statement is made that the detection rate in the present study is low, but the next sentence contradicts this. Recommend editing.
Response: We edited it as recommended.
Reviewer comment: Line 298 needs a citation as it provides information on other coronaviruses sequenced in Brazil
Response: Done.
Reviewer comment: Line 300: recommend using ‘sub-clades’ instead of “clusters”
Response: Done.
Reviewer comment: Lines 301-310: the commentary on related CoVs being detected in closely related hosts, or distantly related hosts, is good. However, this information is not made clear in the results section. Recommend including metadata on phylogenies such as host, sampling location, and year, if such patterns will be assessed in discussion.
Response: Thank you for raising this point. With the new figures generated we addressed these points adding all information available including hosts, sampling location and year of sampling.
Reviewer comment: Line 315-318: The sentence beginning with “In fact…” needs a citation. The data presented in the paper does not at all relate to host switching events for CoVs in Africa and Asia. Where is this information coming from?
Response: We try to make it clearer. The information is from doi: 10.1093/ve/vex012.
Reviewer comment: Commentary on oral vs rectal swab?
Response: We did not understand this question. We discuss oral vc rectal swab in the discussion. Could the receiver be clearer what else should be commented?
Reviewer comment: Line 323-328: this paragraph is unclear and contains grammatical errors. The current phrasing of the sentence which begins on
Response: Thank you. We corrected the whole paragraph.
Reviewer comment: Line 325 suggests that the newly sequenced RdRp sequence may not be a Betacoronavirus. Recommend rephrasing to include that while the sequence confidently branched among Betacoronavirus, it did not fall within an established subgenus.
Response: Thank you, we addressed this point making this section of the text clearer.
Reviewer comment: Line 327: here, like in the results, it is important to include the taxonomic placement of the two sequences it branches closest to.
Response: Thank you, we have added information of taxonomic placement of closely related sequences in the results and discussion section.
Reviewer comment: Line 329: No formal correlation is stated; this statement is unclear.
Response: We corrected this statement. Thank you.
Reviewer comment: Line 400: This study does not directly address spillover risk.
Response: You are right. Corrected.
Reviewer comment: Line 403: It is debatable whether the current study establishes a novel betacoronavirus genus. In figure 2B, it is shown that the sequence generate, representing ‘novel genus’, is in fact very closely related to other betacoronavirus sampled long ago, according to RdRp phylogeny. No new genus classification is provided here, and thus it may be more appropriate to state “a betacoronavirus was detected which does not fall within established genera, and thus may be a member of an unnamed genus”.
Response: We agree with the reviewer and changed the text accordingly.
Reviewer comment: Line 404: The sentence that begins with “Despite” is phrased in such a way that it seems to contradict itself. The absence of SARS-CoV-2 in bats supports that there have been little or no spillback events, but this is not ‘Despite’ the data.
Response: We modified this statement. Thank you.
Reviewer comment: Line 406-411: Like in the introduction, the ideas surrounding environmental conservation efforts are underdeveloped and feel out of place.
Response: We rephrased this sentence.
Reviewer comment: Grammatical and other language feedback throughout
Response: We have revised the whole manuscript. Thank you.
Reviewer comment: Line 70: recommend a comma between “limited” and “highlighting”
Response: Corrected.
Reviewer comment: Line 74: “multi-hosts” should be edited to “multi-host”
Response: Corrected.
Reviewer comment: Line 103: recommend using word “tested” instead of “probed”; stating that you probed bats may confuse reader as probe-capture enrichment has been used to identify viruses in animals.
Response: suggestion accepted.
Reviewer comment: Line 190: “access” and “each” are probably typos; “To determine from which coronavirus genus the…”
Response: suggestion accepted.
Reviewer comment: Line 209: recommend a colon instead of a comma between “families” and “Emballonuridae”
Response: suggestion accepted.
Reviewer comment: Line 216: Ceara can be corrected to Ceará
Response: Corrected.
Reviewer comment: Line 218: remove comma between ‘bats’ and ‘returned’
Response: Corrected.
Reviewer comment: Line 220: “positive results for pan Coronavirus” is awkward phrasing; recommend revisiting.
Response: suggestion accepted
Reviewer comment: Line 224/225: awkward phrasing of sentence. Recommend switching clauses: “A Betacoronavirus was detected in one male …”
Response: suggestion accepted
Reviewer comment: Line 226: recommend removing comma between “two” and “were”
Response: suggestion accepted
Reviewer comment: Line 242 to 252: multiple cases of “genuses”; use “genera”
Response: Corrected.
Reviewer comment: Line 246 and later in results: recommend referring to smaller portions of a phylogenetic tree as a ‘pruned tree’ rather than a ‘Zoom’
Response: suggestion accepted
Reviewer comment: Line 251: “it showed a basal positioning” is vague phrasing for describing the relationship among the major groups of betacoronaviruses. Recommend ommitting.
Response: We agree with the reviewer and omitted this section of the sentence.
Reviewer comment: Line 252: spelling error; edit to “and did not cluster within” or “and did not resolve near”
Response: Corrected.
Reviewer comment: Line 258: possible spelling error; maybe should be “sampled from different regions..”
Response: Corrected.
Reviewer comment: Line 325: grammatical error; switch to “did not cluster”
Response: Corrected.
Reviewer comment: Line 339/340: looks to be a missing word (or words); recommend rephrasing.
Response: Corrected.
Reviewer comment: Line 380: typographical error; “discardedin” should become “discarded in”
Response: Corrected.
Reviewer 2 Report
Comments and Suggestions for Authors
The paper focuses on surveillance among neotropical bats for coronaviruses and influenza viruses. The main aim of the paper was to look for SARS-CoV2 and influenza A, none of which was identified. The study has a small sample size, spread out over various species, locations and 1 year of sampling effort. Mainly alphacoronaviruses were detected and a betacoronavirus. No epidemiological investigation or viral genome recovery was included in the current research.
Since the paper is all about coronaviruses in Neotropical bats - the introduction may be more tailored to include some background on what has been done before, rather than so much that is not directly related to coronaviruses in bats.
Why was there such a focus on detection of SARS-CoV2 in these bats? where is the evidence supporting that this is something that the authors would have found? Rhinolophus are hosts for sarbecoviruses and genetic diversity related to SARS-CoV1 and SARS-CoV2. Human strains of SARS-CoV2 are not merely known to be circulating widely in bats. I have an issue with this point as an important aspect of the study. Similarly, diverse influenza A viruses have been identified in South American bats, but these are not the strains circulating among human populations. The text makes it sound like the authors set out to identify human strains of influenza and SARS-CoV 2 in bats. This is not correct and enflames negative perceptions of bats (particularly in the discussion section). I would suggest that the authors not make such a big deal about aiming to search for these specific viruses among the studied bats as it is baseless. If this was an actual concern, it would be detected via general cov surveillance among bats anyway.
Recently, with bat coronavirus studies, the science has moved past reports of diversity by only reporting small sample sizes, short sequences and short surveillance times. These studies need to be supported by investigating the epidemiology of the infection among natural hosts in an attempt to understand how spillover may occur (even an investigation between the sites sampled in the paper) or then paired by investigating the viral genomes identified in better detail. The study has a small sample size and only short sequences. The very diverse betacoronavirus is even said in the paper to represent a major important finding of the research- why were there no attempts at recovering the complete genome of at least just this betacoronavirus? especially since it is rather rare and its placement in an existing genus or as part of a new genus would be very interesting. This truly would be a major contribution to the research field.
the conclusion also repeats already-established findings from cov literature (no visual symptoms seem among infected bats, tropism, shedding timeframes, evolution etc) with very little input into these aspects from the current study.
Specific issues:
-
Overall the sample sizes for the study is very small, then divided per site and then per species the sample sizes become even smaller. Could the authors comment on the possible population size compositions of the sampled species to maybe account for why so few bats were caught?
-
in addition to the family wide coronavirus assay, did the authors not consider using a similar influenza A virus assay that would detect a wider diversity?
-
Lines 189 and 192: Coronaviridae is a viral family and should always be capitalized and italicized. Please correct its use throughout.
-
Line 206: if genera names are used in the context of taxonomy, they should be in italics and capitalized (Betacoronavirus). If referring to members of a genus they dont have to be in italics, capitalized and can be plural - betacoronaviruses. Similarly line 288 should not be beta coronaviruses but betacoronaviruses. Please correct this throughout the paper.
-
Line 241: Figure 2 is written twice
-
Line 243: genuses is genera or then subgenera
-
Figure 2: the stylized tree is very striking but practically doesnt add much value. One cannot discern what was identified and with what it clusters - particularly among the alphacoronaviruses. There are not that many sequences, the authors can easily show the sequence names in the tree. The need for part B then becomes redundant. Are the sequences in B not also in A? Line 247 “100 best blast hits against the non-redundant NCBI database.” add confusion as there are not 100 sequences in part B. Moreover the figure caption says “Phylogenetic tree reconstructed based on the full RdRp of reference coronavirus genome”- was the entire full RdRp gene used in the tree? no mention of extending the 440 bp region form the sequences identified in the study was mentioned in the materials and methods? is this tree based on a 440bp sequence or a longer one? how can you construct a tree based on unequal sequence lengths?
-
I dont understand the role of figure 2 if there are “higher resolution phylogenetic analysis” figures in figure 3? This seems like an unnecessary amount of extra figures? I also think the figure 3B in line 255 should be 2B?
-
Regarding differences in detection rates for neotropical bats, the authors should consider more carefully looking at the data from the studies cited. Instead of making a generalization overall, perhaps the differences in detection rates can be attributed to seasonality or increased members of certain species included in different studies?
-
Line 350: ‘Visual’ clinical symptoms. What would constitute a clinical symptom from a bat?
Minor issues that can be corrected with proof reading. Mostly issues around nomenclature of coronavirus taxonomy
Author Response
Comments and Suggestions for Authors
Reviewer comment: The paper focuses on surveillance among neotropical bats for coronaviruses and influenza viruses. The main aim of the paper was to look for SARS-CoV2 and influenza A, none of which was identified. The study has a small sample size, spread out over various species, locations and 1 year of sampling effort. Mainly alphacoronaviruses were detected and a betacoronavirus. No epidemiological investigation or viral genome recovery was included in the current research.
Since the paper is all about coronaviruses in Neotropical bats - the introduction may be more tailored to include some background on what has been done before, rather than so much that is not directly related to coronaviruses in bats.
Response: As requested, we modified the introduction adding more background on what has been done before and removed the first paragraph that was not directly connected with bats and coronaviruses.
Reviewer comment: Why was there such a focus on detection of SARS-CoV2 in these bats? where is the evidence supporting that this is something that the authors would have found? Rhinolophus are hosts for sarbecoviruses and genetic diversity related to SARS-CoV1 and SARS-CoV2. Human strains of SARS-CoV2 are not merely known to be circulating widely in bats. I have an issue with this point as an important aspect of the study.
Response: You are correct that Rhinolophus bats are hosts for sarbecoviruses, as we mentioned in the discussion. However, at the onset of the pandemic in Brazil, there was considerable uncertainty regarding the potential behavior of the virus and its spread. Given that SARS-CoV-2 had already been reported to infect various animals from different orders, we deemed it necessary to conduct monitoring efforts.
Reviewer comment: Similarly, diverse influenza A viruses have been identified in South American bats, but these are not the strains circulating among human populations. The text makes it sound like the authors set out to identify human strains of influenza and SARS-CoV 2 in bats. This is not correct and enflames negative perceptions of bats (particularly in the discussion section). I would suggest that the authors not make such a big deal about aiming to search for these specific viruses among the studied bats as it is baseless. If this was an actual concern, it would be detected via general cov surveillance among bats anyway.
Response: While we appreciate your perspective, we respectfully disagree with the notion that monitoring for influenza among bat populations is baseless and we did not understand what general cov surveillance among bats the rewire is referring to?
We believe that surveillance efforts are crucial for understanding the dynamics of virus circulation and potential spillover events. We are aware that influenza A found in bats is not considered a significant concern right now, and only a limited number of research papers demonstrate evidence of bats carrying influenza viruses, but they all emphasize the need of further studies, and the recent one, published in Nature in 2024 (https://doi.org/10.1038/s41467-024-47635-4) suggests that the influenza A(H9N2) virus found in a bat, has features associated with increased risk to humans. Thus, we maintain that it is crucial to monitor wildlife, and publish negative results as well. Therefore, we stand by this aspect of the discussion as it is.
Reviewer comment: Recently, with bat coronavirus studies, the science has moved past reports of diversity by only reporting small sample sizes, short sequences and short surveillance times. These studies need to be supported by investigating the epidemiology of the infection among natural hosts in an attempt to understand how spillover may occur (even an investigation between the sites sampled in the paper) or then paired by investigating the viral genomes identified in better detail. The study has a small sample size and only short sequences. The very diverse betacoronavirus is even said in the paper to represent a major important finding of the research- why were there no attempts at recovering the complete genome of at least just this betacoronavirus? especially since it is rather rare and its placement in an existing genus or as part of a new genus would be very interesting. This truly would be a major contribution to the research field.
Response: Yes, we agree and we acknowledge the limitations of our study. Ideally, complete genome recovery would be optimal; however, we are currently in the process of standardizing protocols for this purpose. We attempted to utilize the Pan Coronavirus panel from Illumina, but unfortunately, we did not achieve success. Presently, we are developing new protocols and anticipate that with further samples, we will be able to sequence the entire genome. However, with the current samples, it is not possible due to the substantial loss of material during the extraction for RT-PCR testing.
As for the low number - as it is further explained, not all captured bats were analyzed. One of the reasons was the financial limitation of our study.
However, when we see for ex some articles about bats and coronavrises in Brazil Bittar et al. 2019, Microbial Ecology tested 119 bats from two regions in Brazil (https://doi.org/10.1007/s00248-019-01391-x)- - Asano et al. 2016 (DOI 10.1186/s12985-016-0569-4) Virology Journal tested 305 animals, Violet‑Lozano et al. 2023 (https://doi.org/10.1007/s42770-022-00878-z) 105 bats tested and Cerri et al. 2023, Microbiolgy Spectrum (10.1128/spectrum.02047-23) tested 47 samples of bats feces.
Reviewer comment: the conclusion also repeats already-established findings from cov literature (no visual symptoms seem among infected bats, tropism, shedding timeframes, evolution etc) with very little input into these aspects from the current study.
Response: We don't believe it's necessary to redefine paradigms in order to publish valuable data. Our study contributes new information on poorly studied coronaviruses in Brazilian bats, including a study conducted in the northeastern states of Brazil. While we acknowledge that some aspects of our conclusions may align with established findings in the coronavirus literature, we believe that reaffirming these points within the context of our study adds to the overall understanding and importance of our findings.
Specific issues:
Reviewer comment: Overall the sample sizes for the study is very small, then divided per site and then per species the sample sizes become even smaller. Could the authors comment on the possible population size compositions of the sampled species to maybe account for why so few bats were caught?
Response: We have added the information to the text to justify the sample size and why some species are more common than others. The field sample size exceeds the number of bats analyzed for viruses in the study. The sample size presented herein pertains to the number of bats analyzed for viruses. However, it would not be logical to include those not tested for viruses. Regarding species, the mist net method employed is more effective for capturing bats of the Phyllostomidae family, thus limiting the number of individuals from other species. Furthermore, the sampled areas were predominantly peri-urban, which selectively filters species, with the most common being from the subfamilies Stenodermatinae, Carolliinae, and Glossophaginae.
Reviewer comment: in addition to the family wide coronavirus assay, did the authors not consider using a similar influenza A virus assay that would detect a wider diversity?
Response: In fact we did it, the RT-PCR for influenza A/SARS-2 is designed to detect all influenza A viruses. We add this information in the text for more clarity.
Reviewer comment: Lines 189 and 192: Coronaviridae is a viral family and should always be capitalized and italicized. Please correct its use throughout.
Response: Thank you. Corrected.
Reviewer comment: Line 206: if genera names are used in the context of taxonomy, they should be in italics and capitalized (Betacoronavirus). If referring to members of a genus they dont have to be in italics, capitalized and can be plural - betacoronaviruses. Similarly line 288 should not be beta coronaviruses but betacoronaviruses. Please correct this throughout the paper.
Response: Thank you. Corrected.
Reviewer comment: Line 241: Figure 2 is written twice
Response:Thank you. Corrected.
Reviewer comment: Line 243: genuses is genera or then subgenera
Response:Thank you. Corrected.
Reviewer comment: Figure 2: the stylized tree is very striking but practically doesnt add much value. One cannot discern what was identified and with what it clusters - particularly among the alphacoronaviruses. There are not that many sequences, the authors can easily show the sequence names in the tree. The need for part B then becomes redundant. Are the sequences in B not also in A? Line 247 “100 best blast hits against the non-redundant NCBI database.” add confusion as there are not 100 sequences in part B.
Response: We thank the reviewer for the comment. We generated new figures with increased tip name sizes for better clarity. Moreover, we splitted figure 3 in figure 3 and 4 to improve clarity and to allow the reader to clearly see the sequence's name and metadata. In fact, there is a substantial amount of alphacoronavirus sequences currently available in databases. For instance, figure 2B is a subsection of a tree built with 70 unique betacoronavirus sequences plus 48 reference sequences. On the other hand, for alphacoronaviruses there are 458 unique sequences plus 48 reference sequences (refer to Suplamentary material). Therefore, it is not trivial to show the phylogenetic context of all sequences recovered.
Reviewer comment: The need for part B then becomes redundant. Are the sequences in B not also in A? Line 247 “100 best blast hits against the non-redundant NCBI database.” add confusion as there are not 100 sequences in part B.
Response: Regarding section A and B, yes, There are many sequences included in part B (most not shown once it is a zoom in a specific clade where the sequence recovered in this study clustered) that are not in part A. Part A is a more general figure to show the specific Coronaviridae genera based on a small representative set of reference sequences from all currently known genera and where the sequences from the study clustered. While Figure 2B and Figure 3 and 4 are excerpts of phylogenetic trees with much higher resolution based on the recovery of more similar sequences to the queries used (sequences generated in this study). Therefore, these figures really show the most closely related sequences available in databases.
Reviewer comment: Line 247 “100 best blast hits against the non-redundant NCBI database.” add confusion as there are not 100 sequences in part B.
Response: We agreed with the reviewer and changed the text accordingly.
Reviewer comment: Moreover the figure caption says “Phylogenetic tree reconstructed based on the full RdRp of reference coronavirus genome”- was the entire full RdRp gene used in the tree? no mention of extending the 440 bp region form the sequences identified in the study was mentioned in the materials and methods? is this tree based on a 440bp sequence or a longer one? how can you construct a tree based on unequal sequence lengths?
Response: Thank you for raising this point, we clarified these points in the new version of the manuscript. First, we used complete RdRp amino acid sequences of reference genomes plus the amino acid sequences recovered from the 440bp fragments generated in this study. There are some recent studies showing that combining longer and shorter sequences can be beneficial to recover more stable phylogenetic trees and positionate shorter fragments into reference trees (https://www.nature.com/articles/s41467-023-39847-x). We used complete and partial aa sequences and were able to recover the expected phylogenetic relationship within the Coronaviridae family and were able to positionate the new sequences within this context. Lastly, for Figure 2B and Figure 3, all were recovered from 440bp nucleotide sequences including the ones extracted from reference genomes.
Reviewer comment: I dont understand the role of figure 2 if there are “higher resolution phylogenetic analysis” figures in figure 3? This seems like an unnecessary amount of extra figures? I also think the figure 3B in line 255 should be 2B?
Response:Thanks for raising this point. Please refer to the previous answers regarding high resolution trees. We corrected Figure 2B accordingly.
Reviewer comment: Regarding differences in detection rates for neotropical bats, the authors should consider more carefully looking at the data from the studies cited. Instead of making a generalization overall, perhaps the differences in detection rates can be attributed to seasonality or increased members of certain species included in different studies?
Response: Thank you for the observation. We agree. Other studies were conducted in different geographical locations, thousands km away. And yes, for sure the differences in detection rates can be attributed to increased members of certain species included in different studies. We add the comment about it.
Reviewer comment: Line 350: ‘Visual’ clinical symptoms. What would constitute a clinical symptom from a bat?
Response: We changed it to "The captured bats showed no visible signs of disease...", because clinical symptoms in bats can include observable abnormalities in behavior, appearance, or physiology. These may include 1) Abnormal flying patterns or difficulty in flight. 2) Visible lesions or wounds on the body. 3) Changes in posture or movement patterns. 4) Respiratory distress, such as rapid or labored breathing. 5) Neurological signs, such as tremors or seizures, etc. These symptoms can indicate potential health issues or infections in bats, including those caused by coronaviruses.
Reviewer comment: Comments on the Quality of English Language
Minor issues that can be corrected with proof reading. Mostly issues around nomenclature of coronavirus taxonomy
Response: We revised the whole manuscript as requested. Thank you.
Reviewer 3 Report
Comments and Suggestions for Authors
Overall the manuscript describe surveillence of bats in 3 municipalities. The manuscript is clear, relevant for the field and presented in a well-structured manner. The manuscript is scientifically sound. The figures and table are appropriate. The conclusions are consistent with the evidence and arguments presented. However, the manuscript can be further improved. Below are a few comments
1) The authors should explain the significance of choosing the 3 municipalities? Are these areas considered eco-tourisms places with possible interaction with humans? Are the areas been recognised as hot spots for viral spill over to other animals?
2) Is there a checklist of biodiversity of bats recorded in the 3 areas or near these areas previously? If yes, discuss the biodiversity of bats in this study in comparison to previous checklist. This can be linked to any previous conservation effort documented.
3) Wantanabe-primers are recognised as one of the widely used pan-coronavirus primers. Please discuss the reason(s) of not including this set of primers. Recently publication shown updated Wantanabe derived pan-coronavirus primers (cited below). The paper was published in April 2021 during this study sampling period. The authors should explain why newer version of pan- coronavirus primers was not adapted?
Holbrook MG, Anthony SJ, Navarrete-Macias I, Bestebroer T, Munster VJ, van Doremalen N. Updated and Validated Pan-Coronavirus PCR Assay to Detect All Coronavirus Genera. Viruses. 2021 Apr 1;13(4):599. doi: 10.3390/v13040599.
4) The authors should include limitation(s) of this study. E.g. no isolation of virus was performed.
Author Response
Comments and Suggestions for Authors
Reviewer comment: Overall the manuscript describe surveillence of bats in 3 municipalities. The manuscript is clear, relevant for the field and presented in a well-structured manner. The manuscript is scientifically sound. The figures and table are appropriate. The conclusions are consistent with the evidence and arguments presented. However, the manuscript can be further improved. Below are a few comments
Response: Thank you.
Reviewer comment: 1) The authors should explain the significance of choosing the 3 municipalities? Are these areas considered eco-tourisms places with possible interaction with humans? Are the areas been recognised as hot spots for viral spill over to other animals?
Response: We added to the text the importance of sampling the study area. In addition to being one of the most important enclaves of moist forest in Ceará, the regions are considered ecotourism sites with potential human interaction.
Reviewer comment: 2) Is there a checklist of biodiversity of bats recorded in the 3 areas or near these areas previously? If yes, discuss the biodiversity of bats in this study in comparison to previous checklist. This can be linked to any previous conservation effort documented.
Response: We added a paragraph discussing the diversity of bats sampled in the study, citing other studies on bat biodiversity in the same study area or nearby regions. Additionally, we discussed the method used and how it impacts the bat biodiversity in this study. This comparison provides valuable insights into the bat fauna of the region and can inform future conservation efforts. “Bat fauna sampling was conducted using mist nets, a method commonly employed in the Neotropical region. Although efficient in capturing bats of the Phyllostomidae family, especially frugivorous species, their effectiveness is reduced for insectivorous bats as these species tend to fly at higher altitudes and avoid mist nets by using echolocation (Fleming 1986; Kunz & Kurta 1988). However, the recent documentation of the first occurrences of Myotis ruber and Molossus pretiosus species in the same region as the present study in Ceará (Santos-Cavalcante et al. 2024), utilizing mist nets, plays a crucial role in expanding our knowledge of bat biodiversity. These new records represent a valuable contribution to the list of known species in the region (Fernandes-Ferreira et al. 2021), providing a more robust foundation for future research and conservation initiatives aimed at protecting bat diversity in the region. Therefore, this study can contribute to the understanding of bat biodiversity and offers valuable insights for research related to the ecology and epidemiology of diseases with the potential for transmission by bats.”
Reviewer comment: 3) Wantanabe-primers are recognised as one of the widely used pan-coronavirus primers. Please discuss the reason(s) of not including this set of primers. Recently publication shown updated Wantanabe derived pan-coronavirus primers (cited below). The paper was published in April 2021 during this study sampling period. The authors should explain why newer version of pan- coronavirus primers was not adapted?
Holbrook MG, Anthony SJ, Navarrete-Macias I, Bestebroer T, Munster VJ, van Doremalen N. Updated and Validated Pan-Coronavirus PCR Assay to Detect All Coronavirus Genera. Viruses. 2021 Apr 1;13(4):599. doi: 10.3390/v13040599.
Response: Thank you for your observation. We utilize primers that have been widely adopted, initially developed in 2005 by Woo et al. and subsequently utilized by Poon et al. (2005) and numerous other researchers, including those from Brazil. This allowed us to compare the same genome fragment across studies. Similar to other researchers (e.g., https://doi.org/10.1038/s41598-018-24407-x; 10.1128/JVI.05838-11), we have successfully detected Alpha, Beta (present study), Gamma, and Delta coronaviruses (other our studies) using these primers. However, we appreciate your recommendation and will consider testing the suggested primers in future studies.
Reviewer comment: 4) The authors should include limitation(s) of this study. E.g. no isolation of virus was performed.
Response: Thank you for your comment. We could not isolate the virus as the samples were collected in the RNAlater solution. We add the paragraph about limitations of our study.
Round 2
Reviewer 1 Report
Comments and Suggestions for Authors
Overall assessment:
Manuscript improved following first round of review. Consistent typographical errors remain throughout the text. A comment from first round - calling the single betacoronavirus sequence a novel genus - was addressed in one section (Results lines 283-288) but not in the discussion (lines 376-382). Supplementary files with complete trees were made accessible. However, this new supplementary material should be renamed ‘supplementary files’ and it should be stated in the main text that they are nexus files. Currently, the onus is on the audience to download each nexus file and visualize each tree themselves (using appropriate software); thus, these files are not figures.
Minor revisions (grammatical or typographical):
line 55: 'coronaviruses' should be edited to 'coronavirus'. 'suggest' should be edited to 'suggests'
line 96: 'Rhinolophus species' could be placed in parentheses to enhance readability and achieve consistency with rest of paragraph
line 210: Remove "and" in "Amino acid and substitution models..."
line 235: recommend substituting "localities" with "sampling sites"
line 256/257: could enhance clarity of the sentence by adding “a” between “detected” and “betacoronavirus”
line 273: “fragment” could be edited to “fragments”; possible spelling error: “bet” should be “best”
line 284: “clustered” should be edited to “cluster”. The reader is directed to Figure 1A, but should be directed to Figure 2A.
line 285: recommend adding “the novel” before “Betacoronavirus sequence” to enhance readability
line 285/286: reverse the words “other” and “two”. Currently reads “.. clustered with high node support to other two coronavirus sequences...”
line 335: recommend substituting “the variety” with “several” for readability
line 336: spelling error- “alphacoronavirues” should be fixed
line 338: recommend editing “the novel” to “a novel”
line 340: spelling error- “Caerá” should be fixed
line 344: recommend rewording the sentence, currently it is disjointed. Placing a comma after “7%” and removing the word “and” would help readability.
line 347: recommend removing “this diffrentes”
line 399: recommend adding the word “which” before “is in line…”
line 410: recommend revisiting sentence structure. Perhaps “Betacoronavirus” could be placed in parentheses, or omitted altogether.
line 451: recommend substituting “what emphasize” with “which emphasizes”
line 479: recommend adding “a single” before the word “Betacoronavirus”
lines 491-494: spelling errors; “phelogenteic” should be fixed
Results
Figure 2 legend (lines 270-282): Recommend directing the reader to the supplementary file containing the entire tree from which 2B was pruned.
line 294: recommend directing reader to supplementary file that demonstrates the clades shown in 3A and 3B are sisters.
Figure 3: Figure 4 nicely illustrates sister clades- could Figure 3 be modified to a similar format, showcasing that the clades currently in 3A and 3B are sister? They could be stacked on top of each other, which would facilitate larger and more readable node labels.
Figure 3 legend (lines 307-315): Recommend rewording the first sentence; currently its meaning is difficult to interpret. “Maximum likelihood phylogenetic trees pruned tree” is tricky wording. Recommend editing “substitution” to “substitutions” when providing unit for branch length.
Figure 4 legend (lines 322-331): Same suggestions as above for Figure 3 legend.
Supplementary figures: It is understood that the nexus format is a suitable way to share the entire genetic trees from which you have pruned the clades shown in main figures. Need to change “phylogenteic" to “phylogenetic” in supplementary captions (if they are to be provided in the submitted manuscript). The nexus files should be referred to as “supplementary files” as they are not figures.
Discussion:
Line 339: the meaning of the final sentence in the first discussion paragraph is unclear. Recommend revisiting. Perhaps “This study represents the first detection and classification of coronaviruses from bats sampled in the Brazilian State Ceará”
Lines 376-382: as outlined in previous round of review, calling the single betacoronavirus sequence a “novel genus” is incorrect. It is more appropriate to say “a betacoronavirus was detected in A. planirostris which does not cluster within established subgenera. Higher resolution phylogenetic analysis indicated that it is closely related to two other coronavirus sequences which had been recovered from A. planirostris and A. lituratus bats captured in southern Brazil over a decade ago.”
Comments on the Quality of English LanguageEnglish is fine. Some spelling and minor grammatical errors to address.
Author Response
Reviewer 1
Comments and Suggestions for Authors
Overall assessment:
Manuscript improved following first round of review. Consistent typographical errors remain throughout the text. A comment from first round - calling the single betacoronavirus sequence a novel genus - was addressed in one section (Results lines 283-288) but not in the discussion (lines 376-382).
REPLY Our fault – corrected that.
Supplementary files with complete trees were made accessible. However, this new supplementary material should be renamed ‘supplementary files’ and it should be stated in the main text that they are nexus files. Currently, the onus is on the audience to download each nexus file and visualize each tree themselves (using appropriate software); thus, these files are not figures.
REPLY. You are right. Corrected.
Minor revisions (grammatical or typographical):
line 55: 'coronaviruses' should be edited to 'coronavirus'. 'suggest' should be edited to 'suggests'
REPLY - Done
line 96: 'Rhinolophus species' could be placed in parentheses to enhance readability and achieve consistency with rest of paragraph
REPLY - Done
line 210: Remove "and" in "Amino acid and substitution models..."
REPLY - Done
line 235: recommend substituting "localities" with "sampling sites"
REPLY - Done
line 256/257: could enhance clarity of the sentence by adding “a” between “detected” and “betacoronavirus”
REPLY - Done
line 273: “fragment” could be edited to “fragments”; possible spelling error: “bet” should be “best”
REPLY - Done
line 284: “clustered” should be edited to “cluster”. The reader is directed to Figure 1A, but should be directed to Figure 2A.
REPLY - Done
line 285: recommend adding “the novel” before “Betacoronavirus sequence” to enhance readability
REPLY - Done
line 285/286: reverse the words “other” and “two”. Currently reads “.. clustered with high node support to other two coronavirus sequences...”
REPLY - Done
line 335: recommend substituting “the variety” with “several” for readability
REPLY - Done
line 336: spelling error- “alphacoronavirues” should be fixed
REPLY - Done
line 338: recommend editing “the novel” to “a novel”
REPLY – Done
line 340: spelling error- “Caerá” should be fixed
REPLY - Done
line 344: recommend rewording the sentence, currently it is disjointed. Placing a comma after “7%” and removing the word “and” would help readability.
REPLY . We reword the sentence - The 7% prevalence of CoVs we observed falls within the range reported in other studies on neotropical bats.
line 347: recommend removing “this diffrentes”
REPLY - Done
line 399: recommend adding the word “which” before “is in line…”
REPLY - Done
line 410: recommend revisiting sentence structure. Perhaps “Betacoronavirus” could be placed in parentheses, or omitted altogether.
REPLY - Done
line 451: recommend substituting “what emphasize” with “which emphasizes”
REPLY - Done
line 479: recommend adding “a single” before the word “Betacoronavirus”
REPLY - Done
lines 491-494: spelling errors; “phelogenteic” should be fixed
REPLY - Done
Results
Figure 2 legend (lines 270-282): Recommend directing the reader to the supplementary file containing the entire tree from which 2B was pruned.
REPLY - Done
line 294: recommend directing reader to supplementary file that demonstrates the clades shown in 3A and 3B are sisters.
REPLY - Done
Figure 3: Figure 4 nicely illustrates sister clades- could Figure 3 be modified to a similar format, showcasing that the clades currently in 3A and 3B are sister? They could be stacked on top of each other, which would facilitate larger and more readable node labels.
REPLY - Done
Figure 3 legend (lines 307-315): Recommend rewording the first sentence; currently its meaning is difficult to interpret. “Maximum likelihood phylogenetic trees pruned tree” is tricky wording. Recommend editing “substitution” to “substitutions” when providing unit for branch length.
REPLY - Done
Figure 4 legend (lines 322-331): Same suggestions as above for Figure 3 legend.
REPLY - Done
Supplementary figures: It is understood that the nexus format is a suitable way to share the entire genetic trees from which you have pruned the clades shown in main figures. Need to change “phylogenteic" to “phylogenetic” in supplementary captions (if they are to be provided in the submitted manuscript). The nexus files should be referred to as “supplementary files” as they are not figures.
REPLY - Done
Discussion:
Line 339: the meaning of the final sentence in the first discussion paragraph is unclear. Recommend revisiting. Perhaps “This study represents the first detection and classification of coronaviruses from bats sampled in the Brazilian State Ceará”
REPLY - Done
Lines 376-382: as outlined in previous round of review, calling the single betacoronavirus sequence a “novel genus” is incorrect. It is more appropriate to say “a betacoronavirus was detected in A. planirostris which does not cluster within established subgenera. Higher resolution phylogenetic analysis indicated that it is closely related to two other coronavirus sequences which had been recovered from A. planirostris and A. lituratus bats captured in southern Brazil over a decade ago.”
REPLY - Done
Comments on the Quality of English Language
English is fine. Some spelling and minor grammatical errors to address.
REPLY – We have reviewed the entire manuscript. Thank you.